# Increasing the Grain Yield and Grain Protein Content of Common Wheat (*Triticum aestivum*) by Introducing Missense Mutations in the *Q* Gene

**DOI:** 10.3390/ijms231810772

**Published:** 2022-09-15

**Authors:** Qing Chen, Zhenru Guo, Xiaoli Shi, Meiqiao Wei, Yazhen Fan, Jing Zhu, Ting Zheng, Yan Wang, Li Kong, Mei Deng, Xinyou Cao, Jirui Wang, Yuming Wei, Qiantao Jiang, Yunfeng Jiang, Guoyue Chen, Youliang Zheng, Pengfei Qi

**Affiliations:** 1State Key Laboratory of Crop Gene Exploration and Utilization in Southwest China, Chengdu 611130, China; 2Triticeae Research Institute, Sichuan Agricultural University, Chengdu 611130, China; 3Crop Research Institute, Shandong Academy of Agricultural Sciences, Jinan 250100, China

**Keywords:** wheat quality, agronomic trait, mutation, breeding

## Abstract

Grain yield (GY) and grain protein content (GPC) are important traits for wheat breeding and production; however, they are usually negatively correlated. The *Q* gene is the most important domestication gene in cultivated wheat because it influences many traits, including GY and GPC. Allelic variations in the *Q* gene may positively affect both GY and GPC. Accordingly, we characterized two new *Q* alleles (*Q^s1^* and *Q^c1^-N8*) obtained through ethyl methanesulfonate-induced mutagenesis. Compared with the wild-type *Q* allele, *Q^s1^* contains a missense mutation in the sequence encoding the first AP_2_ domain, whereas *Q^c1^-N8* has two missense mutations: one in the sequence encoding the second AP_2_ domain and the other in the microRNA172-binding site. The *Q^s1^* allele did not significantly affect GPC or other processing quality parameters, but it adversely affected GY by decreasing the thousand kernel weight and grain number per spike. In contrast, *Q^c1^-N8* positively affected GPC and GY by increasing the thousand kernel weight and grain number per spike. Thus, we generated novel germplasm relevant for wheat breeding. A specific molecular marker was developed to facilitate the use of the *Q^c1^-N8* allele in breeding. Furthermore, our findings provide useful new information for enhancing cereal crops via non-transgenic approaches.

## 1. Introduction

Common wheat (*Triticum aestivum*) is a major food crop that serves as the primary protein source in the human diet. Wheat provides approximately 18% of the calories and 20% of the proteins consumed by humans worldwide [1]. Therefore, grain yield (GY) and grain protein content (GPC) are critical traits to be considered for wheat breeding and production. Because of increased demand driven by population growth and improvements in living conditions, there is an urgent need for wheat varieties with increased GY and GPC.

Nitrogen applications during wheat production are vital for increasing GY and GPC [2,3,4]. In order to produce wheat with a high GY and GPC, farmers tend to apply large amounts of nitrogen fertilizer to wheat fields, which increases cultivation costs and environmental pollution. Breeding to increase the wheat GY and GPC remains a considerable challenge because of the confirmed negative relationship between the two parameters [5,6,7,8].

Wheat flour has unique processing properties that enable it to be used to make diverse end-products. The end-use quality of wheat is significantly influenced by GPC. The unique processing quality of wheat flour depends on the seed storage proteins, especially gliadins and glutenins, which account for 60–80% of the total GPC [9,10]. Gliadins are monomeric compounds that contribute to dough extensibility [11], whereas glutenins, which are polymeric compounds linked by intermolecular disulfide bonds, affect dough elasticity. Glutenins consist of high and low molecular weight glutenin subunits [12,13].

Because the *Q* gene (TraesCS5A02G473800) influences many important traits, including GY, GPC, grain threshability, grain size, spike morphology, rachis fragility, plant height, and flowering time, it plays a major role in wheat domestication, de-domestication and breeding [14,15,16,17,18]. This gene is located on the long arm of chromosome 5A and encodes a member of the APETALA2 (AP_2_) transcription factor family [15,19]. The AP_2_ transcription factors have diverse functions affecting plant development [20]. The *Q* allele originated from a spontaneous mutation to the microRNA172-binding region of the *q* allele [15]. Similarly, the introduction of another point mutation in the microRNA172-binding site of the *Q* allele resulted in the *Q^c1^* allele [17]. The *q*, *Q*, and *Q^c1^* transcription levels are correlated with the number of point mutations in the microRNA172-binding site [15,17]. Compared with the effects of the *Q* allele, *Q^c1^* increases GPC by approximately 60 g kg^−1^, reflecting the value of *Q^c1^* for wheat breeding. However, *Q^c1^* decreases the longitudinal cell size of rachises, resulting in compact spikes and decreases in GY [17]. Missense mutations in the *Q* sequence encoding the AP_2_ domain can lead to decreased spike density [15,21]. These challenges underline the need to identify or generate new *Q* alleles that positively affect GY and GPC.

In this study, we characterized two new *Q* alleles, namely *Q^s1^* and *Q^c1^-N8*, which have a single missense mutation in the sequences encoding the first and second AP_2_ domains, respectively. They were obtained via the chemical treatment of common wheat lines carrying the *Q* and *Q^c1^* alleles, respectively. The effects of *Q^s1^* and *Q^c1^-N8* on the wheat GY and GPC were investigated.

## 2. Results

### 2.1. Phenotype of the Mutant ss1 Carrying the Q^s^^1^ Allele

In order to elevate the effect of missense mutations in the sequence encoding the AP_2_ domain of the Q protein, a mutant carrying the *Q^s1^* allele (*ss1*; sparse spike 1) was isolated from the M_2_ population of the common wheat cultivar ‘Shumai482′ (Figure 1). The *ss1* plants produced a speltoid-like spike (Figure 2a). In contrast to the *Q* allele (GenBank No. KX580301.2), *Q^s1^* has a missense mutation (GenBank No. OK041024) in the sequence encoding the first AP_2_ domain (Figure 3 and Appendix A). Compared with the WT control (Figure 1), *ss1* plants were taller (Figure 2b, Table 1) and had a longer main spike (Figure 2a, Table 1) but a lower spike density (Table 1).

Regarding the examined yield-related traits, the spikelet number per the main spike, grain number per the main spike, thousand kernel weight (Table 1), and grain width (Figure 2d,e) were lower for *ss1* than for the WT control. In contrast, the grain length (Figure 2c,f) was greater for *ss1* than for the WT control. Notably, there was no significant difference in the productive tiller number between the *ss1* and WT plants (Table 1). Therefore, GY was significantly lower (*p* < 0.01) for *ss1* (0.46 kg m^−2^) than for the WT control (0.58 kg m^−2^) in the 2018–2019 growing season. Under our experimental conditions, the *Q^s1^* allele decreased GY by 20.6% by decreasing the thousand kernel weight and the grain number per spike.

An analysis of the processing quality traits revealed a lack of a significant difference between the *ss1* and WT plants in terms of GPC, wet gluten content, gluten index, Zeleny sedimentation value, water absorption, development time, stability time, and loaf volume (Table 2; Figure 4).

### 2.2. Characterization of the Mutant NS8 Carrying the Q^c1^-N8 Allele

Because a missense mutation in the sequence encoding the AP_2_ domain can decrease the spike density without altering processing quality parameters, the normal-spike mutant *NS*8 containing the *Q^c1^-N8* allele was isolated from the M_2_ population of the mutant *S-Cp*1-1. The *NS*8 plants had a normal spike, which was similar to the ‘Shumai482′ spike (Figure 5a and Appendix A). Compared with the *Q* allele sequence, *Q^c1^-N8* contains two missense mutations (GenBank No. OK041023), with one in the sequence encoding the second AP_2_ domain and the other in the microRNA172-binding region (Figure 3 and Appendix A). When compared with the WT control (plants with only the *Q* allele; Figure 1), the mutant *NS*8 plants were shorter (Figure 5b, Figure 6b, and Appendix A) but had a similar spike length (Figure 5a, Figure 6d, and Appendix A) and spike density (Figure 6f and Appendix A).

Of the yield-related traits, the grain number per spike (Figure 6a and Appendix A), thousand kernel weight (Figure 5e and Appendix A), grain width (Figure 5d,g and Appendix A), and grain length (Figure 5c,h and Appendix A), were greater for the mutant *NS*8 than for the WT control. However, there were no significant differences in the spikelet number per the main spike (Figure 6e and Appendix A) and productive tiller number (Figure 6c and Appendix A) between the *NS*8 and WT plants. As expected, the *Q^c1^-N8* allele positively affected GPC (Figure 5f and Appendix A) and GY by increasing the thousand kernel weight and grain number per spike.

In order to facilitate the use of the *Q^c1^-N8* allele in breeding, the missense mutation in the sequence encoding the second AP_2_ domain was converted to a CAPS marker (Figure 7). The CAPS primer pair (N8-CAPS-F1 + N8-CAPS-R1; Table 3) amplified a 300 bp fragment from the genomic DNA of *NS*8 and WT. The 300 bp fragment of WT could be digested into 210 bp and 90 bp bands. In contrast, the amplicons of *NS*8 remained undigested.

## 3. Discussion

Because GY and GPC determine the profitability of wheat production, they are the primary traits upon which wheat breeders and growers focus. More specifically, GY is critical for ensuring food security, especially in many developing countries, while GPC is a crucial index for assessing the nutritional and unique processing quality of wheat [22]. Increasing both GY and GPC is important to improving the availability of high-quality food and, by extension, the living standards of humans. As an essential macronutrient for plants, nitrogen is crucial for establishing a balance between GY and GPC during wheat production [3,4,23]. Accordingly, the most frequent cultivation practice used by farmers to increase GY and GPC is the application of nitrogen fertilizers [4,24], but this leads to increased cultivation costs, decreased nitrogen use efficiency, and increased environmental pollution. Moreover, after incremental additions of nitrogen fertilizer, GPC reaches a maximum and then remains constant [24]. In this study, we created a new *Q* allele (i.e., *Q^c1^-N8*), which breaks the negative relationship between GY and GPC and can synchronously increase the wheat GY and GPC. Thus, *Q^c1^-N8* may be useful for breeding more profitable wheat varieties.

‘Shumai482′ is an elite commercial wheat cultivar with a relatively high GY and GPC. The *Q^c1^-N8* allele in the ‘Shumai482′ genetic background can increase both GY and GPC. Notably, *Q^c1^-N8* decreased the plant height (Figure 5b, Figure 6b and Appendix A), thereby enhancing lodging resistance. Lodging is still a major factor limiting global wheat production, especially in regions with heavy rain and strong winds, because it leads to serious decreases in GY. We are currently assessing the breeding value of *Q^c1^-N8* in multiple environments, in diverse genetic backgrounds, and in field plot experiments.

As important parts of the Q protein, the AP_2_ domains are critical for DNA binding and for physical interaction with other proteins. Modifying the amino acids in the AP_2_ domains of the Q protein may reverse the unfavorable agronomic traits of the mutant *S-Cp*1-1 carrying the *Q^c1^* allele [17] by affecting the expression of downstream genes and the interaction between Q and other proteins. For example, the transcription factor TaLAX1 physically interacts with Q to antagonistically regulate grain threshability and spike morphology [25]. However, most downstream genes of Q for regulating agronomic traits remain unknown. The mutation in the *Q^s1^* allele results in a single amino acid change in the first AP_2_ domain (Figure 3 and Appendix A), which negatively affects the thousand kernel weight (Table 1). The overexpressed *Q^c1^* allele has a missense mutation in the mircoRNA172-binding site (Figure 3 and Appendix A) that increases the thousand kernel weight [17]. It is likely that at least some missense mutations in the sequences encoding the AP_2_ domains and those in the microRNA172-binding site have the opposite effect on the thousand kernel weight. The *Q^c1^-N8* allele has two missense mutations, with one in the sequence encoding the AP_2_ domain and the other in the microRNA172-binding site (Figure 3 and Appendix A); this allele is associated with an increase in the thousand kernel weight (Figure 5e and Appendix A). Therefore, the opposite effects of the two-point mutations in the *Q^c1^-N8* allele are relatively well balanced to increase the thousand kernel weight. However, further increases in GY require the creation of new alleles with mutation(s) beyond the AP_2_ domain-encoding sequences; four previously reported *Q^c^* alleles (i.e., *Q^c1^*–*Q^c4^*) [17] might be useful for generating new alleles.

Common wheat is a hexaploid species (AABBDD; 2n = 6x = 42) that contains three homologous genomes (i.e., A, B, and D). An earlier study revealed the dosage effect of the *Q* gene in wheat [26]. To date, only the *Q* gene copy in the A genome has been optimized. To further improve GY and GPC, the stepwise optimization of the *Q* copies in the B and D genomes of common wheat and related species is ongoing.

Durum wheat (*Triticum turgidum* ssp. *durum*) is a tetraploid species (AABB; 2n = 4x = 28) and is the main and preferred raw material for pasta production [27]. The GPC is a determining factor influencing durum wheat quality, and grains with a high GPC tend to produce good cooking quality pasta [27,28,29]. The *Q^c1^-N8* allele, which is located in the A genome, may be useful for durum wheat breeding.

In addition to the *Q* gene, many other plant genes include microRNA-binding sites, including some genes encoding a conserved AP_2_ domain. The directed evolution of these genes via the introduction of point mutations in their microRNA-binding sites and other domain-encoding sequences may be an efficient and effective way to ensure global food security.

Increases in GY and GPC are also required for other major cereal crops, such as rice, maize, barley, sorghum, and foxtail millet, all of which carry *Q* gene orthologs and homologs [20,30,31,32,33,34]. Moreover, these orthologous and homologous genes seem to have conserved functions among cereals [31,33,34]. Similar to the allele development in this study (i.e., *Q* allele to *Q^c1^* and then to *Q^c1^-N8*), elite alleles for the *Q* gene orthologs and homologs can be created by the stepwise optimization of their expression (e.g., by introducing point mutations in the mircoRNA172-binding site or in other elements) and by enhancement of the activities of the encoded proteins (e.g., by introducing point mutations in the sequences encoding the AP_2_ domains or other domains), affecting specific downstream gene(s) and interacting protein(s). As they are applied in breeding programs involving non-transgenic methods, these attributes illustrate the utility of creating a set of elite alleles of the *Q* gene orthologs and homologs to increase the GY and GPC of cereal crops.

## 4. Materials and Methods

### 4.1. Plant Materials and Growth Conditions

The seeds of common wheat cultivar ‘Shumai482′ (*Q* allele) and its compact-spike mutant *S-Cp*1-1 (*Q^c1^* allele) [17] were treated with 0.8% and 0.4% ethyl methanesulfonate (Catalog number: M0880-100G; Sigma-Aldrich, St Louis, MO, USA), respectively. Seeds from the leading spikes of the M_1_ plants were harvested and sown to generate the M_2_ population. The mutant *ss1* (sparse spike 1) was obtained from the M_2_ population of ‘Shumai482′. The mutant *NS*8 (normal spike 8) was isolated from the M_2_ population of *S-Cp*1-1. The *Q* genes of *ss1* (*Q^s1^* allele) and *NS*8 (*Q^c1^-N8* allele) were sequenced.

The mutants were backcrossed with ‘Shumai482′ to assess the effects of *Q^s1^* and *Q^c1^-N8* on agronomic traits and processing quality parameters. Ten BC_2_F_3_ homozygous lines (five with the *Q* allele and five with the *Q^s1^* allele) and 10 BC_2_F_4_ homozygous lines (five with the *Q* allele and five with the *Q^s1^* allele) (Figure 1) were grown at the experimental farm of Sichuan Agricultural University in Wenjiang (30°43′16″ N, 103°52′15″ E) during the 2018–2019 and 2019–2020 wheat growing seasons, respectively. Field trials were performed using a randomized block design. Each line was cultivated in a 2 m × 3 m area, with a row spacing of 20 cm *×* 5 cm. The BC_1_F_2_ and BC_2_F_2_ plants carrying *Q* or *Q^c1^-N8* (Figure 1) were grown with a row spacing of 20 cm *×* 10 cm in Wenjiang during the 2020–2021 and 2021–2022 growing seasons, respectively. A nitrogen:phosphorous:potassium (15:15:15) compound fertilizer was applied before sowing (450 kg per hectare).

At the GS87 growth stage [35], agronomic traits, including plant height (cm), spike length (cm), spikelet number per spike, grain number per spike, and productive tiller number, were recorded. Spike density was calculated as the ratio of the spike length to the spikelet number per spike. For the BC_2_F_3_ and BC_2_F_4_ homozygous lines with the *Q* or *Q^s1^* allele, 20 representative plants of each line were examined. For the homozygous BC_1_F_2_ and BC_2_F_2_ plants carrying the *Q* or *Q^c1^-N8* allele, the agronomic traits of 20–30 plants were also evaluated.

After harvesting samples and drying them under the sun at approximately 35 °C to a constant weight, the thousand kernel weight (g), grain length (mm), and grain width (mm) were determined. For each BC_2_F_3_ and BC_2_F_4_ homozygous line with the *Q* or *Q^s1^* allele, the thousand kernel weight was measured by randomly selecting 1000 seeds. For the BC_1_F_2_ and BC_2_F_2_ plants carrying the *Q* or *Q^c1^-N8* allele, the thousand kernel weight was measured on the basis of 200 randomly selected mature seeds. In order to measure the grain length and width, 100 randomly selected seeds were scanned using the Epson Eu-88 A3 Transparency Unit (Seiko Epson, Nagano, Japan). The resulting images were analyzed using the WinSEEDLE Analysis System (Regent Instruments, QC, Canada).

### 4.2. Gene Cloning

Young leaves collected from individual plants at the GS13 growth stage [35] were ground to a fine powder in liquid nitrogen. Genomic DNA and total RNA were extracted from the ground materials using Plant DNA/RNA extraction kits, respectively (Catalog numbers: DN32-100 and RN33050; Biofit, Chengdu, China). First-strand cDNA was synthesized using the Prime Script™ 1st Strand cDNA Synthesis Kit (Catalog number: 6110A; Takara, Dalian, China). All kits were used as recommended by the manufacturers.

The *Q* cDNA and genomic DNA sequences of the mutants *ss1* and *NS*8 were cloned and sequenced. The PCR amplifications were completed in a 50 µL volume consisting of genomic DNA or cDNA, 200 µM dNTPs, 10 µM each primer, 1 U Phanta Max Super-Fidelity DNA Polymerase (Catalog number: P505-d1/d2/d3; Vazyme, Nanjing, China), and 25 µL 2× supplied buffer (with Mg^2+^). The PCR was performed using the Mastercycler Pro thermal cycler (Eppendorf, Hamburg, Germany) with the following program: 95 °C for 5 min; 35 cycles of 95 °C for 45 s, 60–68 °C for 30 s, and 72 °C for 2 min; 10 min at 72 °C. The PCR products were separated on a 1.5% agarose gel (Catalog number: 5260; Takara). The target fragments were purified using the FastPure Gel DNA Extraction Mini Kit (Catalog number: DC301-01; Vazyme) and then inserted into the pCE2 TA/Blunt-Zero vector using the 5 min TA/Blunt-Zero Cloning Kit (Catalog number: C602-01; Vazyme). Positive colonies were sequenced by Sangon Biotech (Chengdu, China). The cloning and sequencing experiments were repeated at least three times. Sequences were analyzed using DNAMAN (version 8) (Lynnon Biosoft, San Ramon, CA, USA). The primers used are listed in Table 3.

### 4.3. Genotyping for Q^c1^-N8

Genomic DNA extracted from individual plants in the BC_1_F_2_ and BC_2_F_2_ population of the mutant *NS*8 was used as the PCR template. The QF7 + QR7 primer pair (Table 3) flanking the microRNA172-binding site was used. The PCR amplifications were performed as described above. The PCR products were sequenced to determine the presence/absence of *Q^c1^-N8*.

### 4.4. Development of CAPS Marker for Q^c1^-N8

The point mutation in the sequence encoding the second AP_2_ domain of the *Q^c1^-N8* allele was converted to the CAPS (Cleaved Amplified Polymorphic Sequence) marker by using DNAMAN (version 8). This CAPS marker (N8-CAPS-F1 + N8-CAPS-R1; Table 3) was tested in the BC_1_F_2_ and BC_2_F_2_ populations of the mutant *NS*8. The PCR amplifications were performed as described above. The PCR products (about 5 µg) were digested with five units of the restriction enzyme *Bbv*I (Catalog number: R0173S; New England Biolabs, Ipswich, MA, USA) along with given buffer at 37 °C for 120 min. The digested fragments were separated on a 1.5% agarose gel.

### 4.5. Processing Quality Analysis

Mature grains were dried under the sun, cleaned, and stored at room temperature for 2 months. The GPC (dry weight) was measured as described by [36]. In order to assess the effect of *Q^s1^* on processing quality, the moisture content of the *ss1* (Figure 1) and wild-type (WT) grains were adjusted to 16.5% before the samples were milled using the CD1 Laboratory Mill (CHOPIN Technologies, Villeneuve-la-Garenne Cedex, France). The Zeleny sedimentation value, wet gluten content, gluten index, and dough rheological properties were determined as previously described [36]. A farinograph (Brabender GmbH & Co., Duisburg, Germany) was used to determine the rheological properties.

A baking test was performed according to a slightly modified version of AACC method 10.09-01 [37]. Specifically, a standard rapid mix test involving 50 g flour (14% moisture content) was conducted. There were two loaves of bread per flour sample. The loaf volume was determined using the BVM6630 volume meter (Pertern, Stockholm, Sweden), as described by the manufacturer.

### 4.6. Statistical Analysis

All data were calculated using Excel 2010 (Microsoft, Redmond, WA, USA). The significance of the differences in the mean values for the agronomic traits and processing quality parameters between the WT and mutant samples was determined according to Student’s *t*-test implemented in the Data Processing System (DPS) software (version 18.10) (Zhejiang University, Hangzhou, China) [38]. The DPS software was also used to perform an analysis of variance.

## 5. Conclusions

In this study, we characterized two new *Q* alleles (*Q^s1^* and *Q^c1^-N8*) and demonstrated that the negative correlation between GY and GPC of wheat could be broken by *Q^c1^-N8* via a non-transgenic approach. *Q^c1^-N8* synchronously increases the wheat GY and GPC, and a specific molecular marker was developed to facilitate its use in breeding. It is possible to further increase both GY and GPC by progressively optimizing the *Q* gene in the three genomes of wheat.

## Figures and Tables

**Figure 1 ijms-23-10772-f001:**
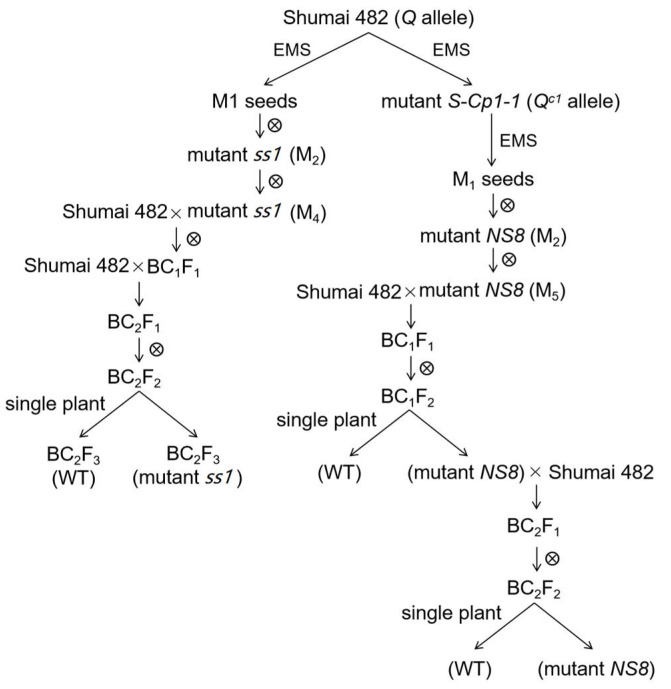
Outline of the generation of the mutants *ss1* and *NS*8 in the *T. aestivum* cv. ‘Shumai 482′ genetic background.

**Figure 2 ijms-23-10772-f002:**
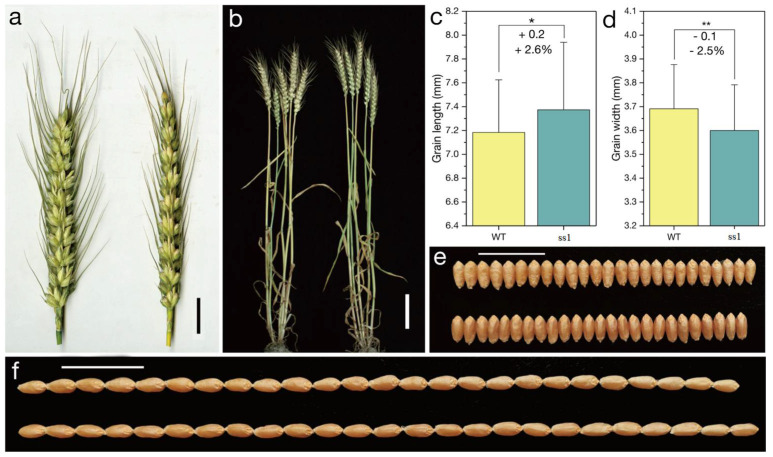
Phenotype of the mutant *ss1*. (**a**) Spikes of the wild-type (WT) (**left**) and *ss1* (**right**) plants. Scale bar, 2 cm. (**b**) WT (**left**) and *ss1* (**right**) plants. Scale bar, 10 cm. Comparisons of grain length (**c**) and grain width (**d**). **, *p* < 0.01; *, *p* < 0.05. Data are presented as the mean ± standard deviation. ‘+’ and ‘−’ indicate more and less than the WT control, respectively. ‘+0.2′ in panel (**c**) indicates the *ss1* grain was 0.2 mm longer than the WT grain (on average). ‘+ 2.6%’ in panel (**c**) indicates *Q^s1^* increased the grain length by 2.6% (on average). (**e**) Kernel width of the WT (**upper**) and *ss1* (**lower**) samples. Scale bar, 2 cm. (**f**) Kernel length of the WT (**upper**) and *ss1* (**lower**) samples. Scale bar, 2 cm.

**Figure 3 ijms-23-10772-f003:**
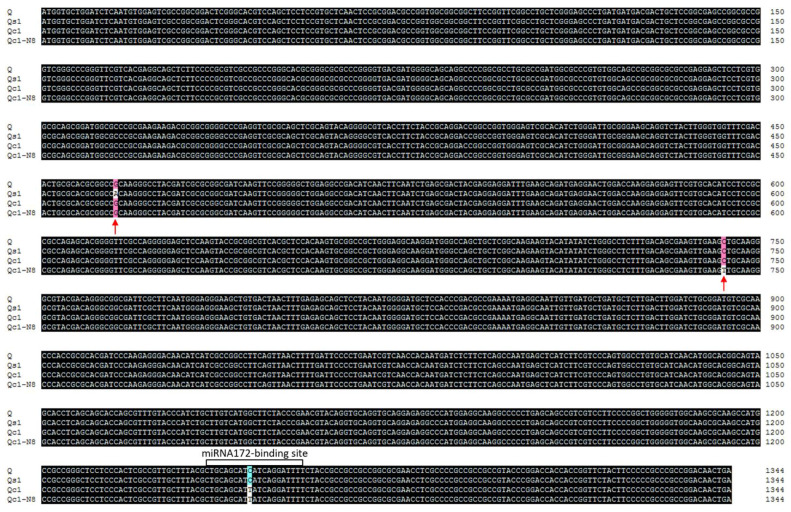
Alignment of the *Q* (GenBank No. KX580301.2), *Q^s1^* (OK041024), *Q^c1^* (KX580302.2), and *Q^c1^-N8* (OK041023) open reading frames. The red arrows indicate the two specific missense mutations in the *Q^s1^* and *Q^c1^-N8* alleles. The microRNA172-binding site is boxed.

**Figure 4 ijms-23-10772-f004:**
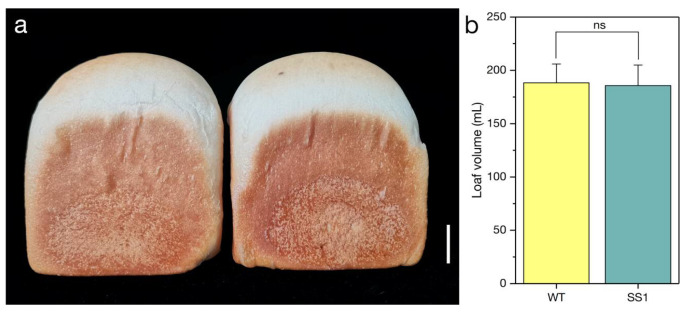
*Q^s1^* has no effect on the bread loaf volume. (**a**) Comparison of the intact *ss1* (**left**) and wild-type (WT) (**right**) loaves. Scale bar, 2 cm. (**b**) Comparison of the *ss1* and WT loaf volumes. ns, not significant.

**Figure 5 ijms-23-10772-f005:**
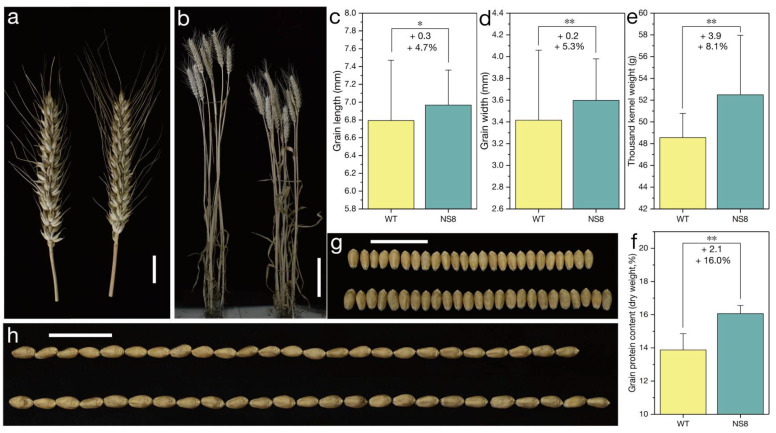
Phenotype of the mutant *NS*8 during the 2020–2021 growing season. (**a**) Spikes of the wild-type (WT) (**left**) and *NS*8 (**right**) plants. Scale bar, 2 cm. (**b**) WT (**left**) and *NS*8 (**right**) plants. Scale bar, 10 cm. Comparisons of the grain length (**c**), grain width (**d**), thousand kernel weight (**e**), and grain protein contents (**f**) of the WT and *NS*8 samples. **, *p* < 0.01; *, *p* < 0.05. Data are presented as the mean ± standard deviation. ‘+’ indicates more than the WT control. ‘+0.3′ in panel (**c**) indicates the *NS*8 grain was 0.3 mm longer than the WT grain (on average). ‘+4.7%’ in panel (**c**) indicates that *Q^c1^-N8* increased the grain length by 4.7% (on average). (**g**) Kernel width of the WT (**upper**) and *NS*8 (**lower**) samples. Scale bar, 2 cm. (**h**) Kernel length of the WT (**upper**) and *NS*8 (**lower**) samples. Scale bar, 2 cm.

**Figure 6 ijms-23-10772-f006:**
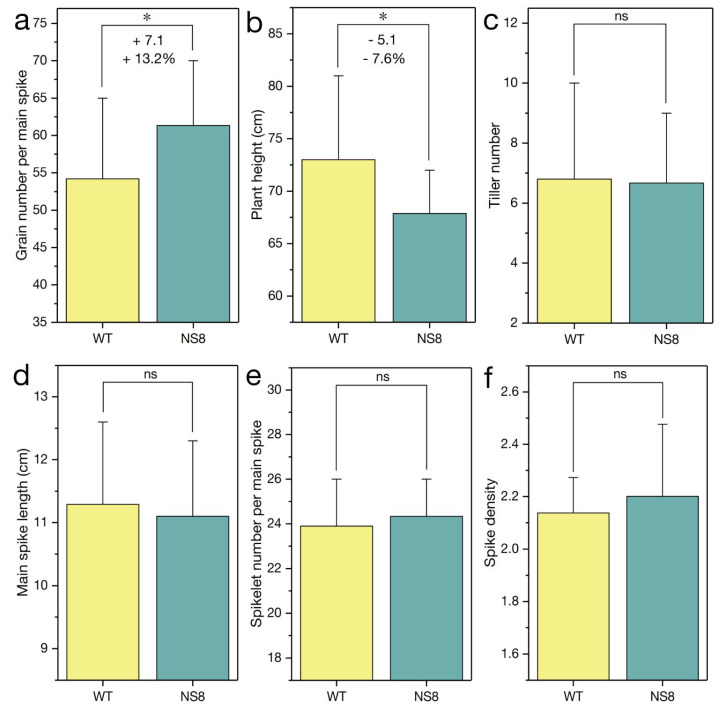
Effect of *Q^c1^-N8* on agronomic traits during the 2020–2021 growing season. The subfigures (**a**–**f**) are the comparisons of the grain number per main spike, plant height, tiller number, main spike length, spikelet number per main spike, and spike density of the WT and *NS*8 samples, respectively. The data in the bar graphs are presented as the mean ± standard deviation. *, *p* < 0.05; ns, not significant. ‘+’ and ‘−’ indicate more and less than the wild-type (WT) control, respectively. ‘+7.1′ in panel (**a**) indicates *NS*8 plants had 7.1 more grains per main spike than the WT plants (on average). ‘+13.2%’ in panel (**a**) indicates that *Q^c1^-N8* increased the grain number per main spike by 13.2% (on average).

**Figure 7 ijms-23-10772-f007:**
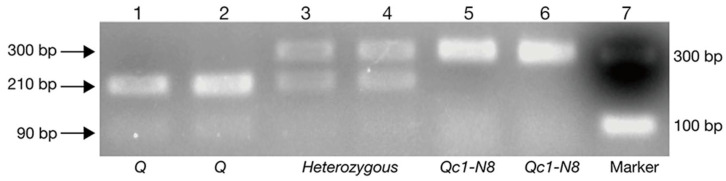
Gel separation of digested and undigested PCR products amplified by using the primer pair N8-CAPS-F1 + N8-CAPS-R1. Lanes 1 and 2 are the digested results of the *Q* allele. Lanes 3 and 4 are the digested results of the *Q*/*Q^c1^-N8* alleles. Lanes 5 and 6 are the undigested results of the *Q^c1^-N8* allele. Lane 7 is the DNA marker.

**Table 1 ijms-23-10772-t001:** Comparison of the agronomic traits of the mutant *ss1* and the wild-type (WT) control.

Traits	Growing Season	*ss1*	WT	E	G	E × G
Plant height (cm)	2018–2019 2019–2020	83.41 ± 4.05 ** 71.82 ± 2.30 **	79.44 ± 3.09 67.11 ± 2.00	205.087 **	24.459 **	0.534
Spike length (cm)	2018–2019 2019–2020	14.65 ± 0.68 * 12.88 ± 0.80 **	14.06 ± 0.75 11.87 ± 0.46	123.331 **	15.408 **	2.680
Spikelet number per main spike	2018–2019 2019–2020	21.00 ± 1.15 ** 19.93 ± 0.70 **	22.94 ± 1.44 21.43 ± 1.45	15.038 **	24.858 **	0.110
Spike density	2018–2019 2019–2020	1.43 ± 0.07 ** 1.55 ± 0.10 **	1.63 ± 0.09 1.80 ± 0.08	45.434 **	84.944 **	13.78
Grain number per main spike	2018–2019 2019–2020	58.81 ± 5.83 * 50.43 ± 4.14 *	65.13 ± 8.23 54.64 ± 4.25	43.376 **	8.860 **	0.044
Thousand kernel weight (g)	2018–2019 2019–2020	44.08 ± 1.71 * 48.76 ± 1.25 *	46.13 ± 0.43 51.90 ± 1.67	50.427 **	13.780 **	0.216
Tiller number	2018–2019 2019–2020	5.30 ± 1.03 3.85 ± 0.74	5.25 ± 1.33 4.25 ± 0.79	27.478 **	0.730	1.207

**, *p* < 0.01; *, *p* < 0.05; E, environment; G, genotype; E × G, interaction between the environment and genotype. Data are presented as the mean ± standard deviation.

**Table 2 ijms-23-10772-t002:** Comparison of the processing quality parameters of the mutant *ss1* and the wild-type (WT) control.

Trails	Growing Season	*ss1*	WT	E	G	E × G
Grain protein content (%; dry weight)	2018–2019 2019–2020	13.12 ± 0.43 14.53 ± 0.60	13.45 ± 0.36 15.16 ± 0.76	30.313 **	2.562	0.323
Zeleny sedimentation value (mL)	2018–2019 2019–2020	32.65 ± 3.64 21.54 ± 2.51	27.73 ± 4.20 23.88 ± 1.98	29.114 **	0.301	4.315
Wet gluten content (%)	2018–2019 2019–2020	24.72 ± 1.60 27.21 ± 1.49	24.98 ± 3.06 29.66 ± 2.05	14.645 **	0.570	2.996
Gluten index (%)	2018–2019 2019–2020	92.87 ± 2.59 63.37 ± 9.02	87.32 ± 8.94 65.04 ± 6.69	65.599 **	0.004	0.315
Water absorption (%)	2018–2019 2019–2020	50.36 ± 0.71 58.74 ± 1.12	50.88 ± 1.67 58.28 ± 0.45	428.875 **	0.783	0.077
Development time (s)	2018–2019 2019–2020	78.00 ± 13.22 61.50 ± 10.71	77.67 ± 9.14 68.40 ± 10.46	7.440 *	0.365	2.875
Stability time (s)	2018–2019 2019–2020	260.86 ± 121.40 193.20 ± 56.34	203.83 ± 66.50 230.40 ± 54.54	0.394	0.125	1.560

**, *p* < 0.01; *, *p* < 0.05; E, environment; G, genotype; E × G, interaction between the environment and genotype. Data are presented as the mean ± standard deviation.

**Table 3 ijms-23-10772-t003:** Primers used in this study.

Primers Name	Sequences (5′-3′)	Reference	Objective
AP2startF	ATGGTGCTGGATCTCAATGTGGAGTCGCCGGCGGA	[15]	Cloning of the genomic DNA sequence of *Q* gene
AP2.8R	CGCGGCCAAATCGGGGCAAAGGAATTCAAACGA	[15]	
AP2.2-1F	ATCTTAGCTGTATGGGCTCGTG	This study	
AP2.2-1R	TCAACGGAGATAGGGGTGTG	This study	
AP2.2-2F	AGGCTCCACATAAGTATATGATCGAGTC	This study	
AP2.2-2R	CTTAATTTCAGGAACGAACTTGTCG	This study	
AP2.16F	CTGCTTGGTGCGCTGCTCCACCAGCTTACTGAAA	[15]	
AP45.1R	CAGAAGGCCCAACGGTTAACGCAACAATGGC	[15]	
Q-mRNA-F2-123	TCGGAGATGGTGCTGGAT	This study	Cloning the full open reading frame of *Q* gene
Q-mRNA-R1-1479	GCCAGCTTCAGTTGTCCG	This study	
QF7	GACCAGCCAGTAGTGTCACC	This study	Genotyping of *Q^c1^-N8* allele
QR7	TCTTGCAGTTCCATCCGTCC	This study	
N8-CAPS-F1	ACTAGAGTGAGTGAGCAAAGATT	This study	CAPS marker for *Q^c1^-N8* allele
N8-CAPS-R1	AGCAACTGTTAGGCTCCACATAA	This study	

## Data Availability

All data supporting the findings of this study are available within this article and the Appendix A published online.

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
