# Peer review of "Increasing the Grain Yield and Grain Protein Content of Common Wheat (Triticum aestivum) by Introducing Missense Mutations in the Q Gene"

_ijms, 2022, doi:10.3390/ijms231810772_

Round 1
Reviewer 1 Report
The Q gene is one of the most important domestication gene in wheat, and many researches focus on Q alleles in regulating plant architecture, ie. spike length and density. This manuscript characterized two new Q alleles (Qs1 and Qc1-N8) via EMS-induced mutagenesis, and found that Qs1 allele did not significantly affect grain protein content (GPC), but it adversely affected grain yield (GY); Qc1-N8 positively affected GPC and GY by increasing the thousand kernel weight and grain number per spike. Furthermore, authors generated novel germplasms possibly useful in wheat breeding, and developed a specific molecular marker for screening the Qc1-N8 allele in breeding. Overall, this manuscript is well written and provided available information for Q alleles function and application. I have the following concerns and comments for the authors to consider revising the manuscript.
1. Figure 3 and 4 seem a bit repetitive. Figure 3 or 4 can be moved to the supplementary section.
2. The AP2 domains are important DNA binding domains. So the mutations may enhance or repress downstream genes expression. Possible functional mechanisms of these two alleles need to be discussed in detailed.
Author Response
- Figure 3 and 4 seem a bit repetitive. Figure 3 or 4 can be moved to the supplementary section.
Figure 4 was moved the supplementary section as Figure S1.
- The AP2 domains are important DNA binding domains. So the mutations may enhance or repress downstream genes expression. Possible functional mechanisms of these two alleles need to be discussed in detailed.
Line 198: As important parts of the Q protein, the AP2 domains are critical for DNA binding and for physical interaction with other proteins. Modifying the amino acids in the AP2 domains of the Q protein may reverse the unfavorable agronomic traits of the mutant S-Cp1-1 carrying the Qc1 allele [17], by affecting the expression of downstream genes and the interaction between Q and other proteins. For example, the transcription factor TaLAX1 physically interacts with Q to antagonistically regulate grain threshability and spike morphology [25]. However, the most downstream genes of Q for regulating agronomic traits remains unknown.
Reviewer 2 Report
In this manuscript, authors have reported the mutation in the Q gene. The overall manuscript is well-written. But I have some concerns related to this as follows:
1-In the introduction section details regarding the Q gene are lacking. Please add some useful information related to the Q gene.
2-Overall, the research was conducted well, and good data was collected from the experiment, however, the authors need to justify the novelty of the current study.
3- please provide the Location id of the Q gene.
4-In line 246: The mutant SS1 (sparse spike 1) should be written in small letters. Please check the whole manuscript carefully.
5- Authors have used different chemicals/reagents for a different purposes, please provide their catalog numbers where applicable.
6- Authors used a baking test with slight modifications but they did not mention these modifications.
7- Please add a conclusion section with strong future recommendations.
Author Response
- In the introduction section details regarding the Q gene are lacking. Please add some useful information related to the Q
Line 52: Because the Q gene (TraesCS5A02G473800) influences many important traits, including GY, GPC, grain threshability, grain size, spike morphology, rachis fragility, plant height and flowering time, it plays a major role in wheat domestication, de-domestication and breeding [14-18].
- Overall, the research was conducted well, and good data was collected from the experiment, however, the authors need to justify the novelty of the current study.
Line 41: To produce wheat with a high GY and GPC, farmers tend to apply large amounts of nitrogen fertilizer to wheat fields, which increases cultivation costs and environmental pollution. Breeding to increase the wheat GY and GPC remains a considerable challenge because of the confirmed negative relationship between the two parameters [5-8].
Line 246: In this study, we characterized two new Q alleles (Qs1 and Qc1-N8), and demonstrated that the negative correlation between GY and GPC of wheat can be broken by Qc1-N8 via a non-transgenic approach. Qc1-N8 synchronously increases the wheat GY and GPC, and a specific molecular marker was developed to facilitate its use in breeding. It is possible to further increase both GY and GPC by progressively optimizing of the Q gene.
Therefore, the most important finding of this study is the generation of the Qc1-N8 allele through ethyl methanesulfonate-induced mutagenesis. It is the first report of a gene/allele that can simultaneously increase both GY and GPC in wheat. Notably, it is possible to further increase both GY and GPC by progressively optimizing of the Q gene in the future.
- Please provide the Location id of the Q
Line 52: The ID of the Q gene is TraesCS5A02G473800.
- In line 246: The mutant SS1 (sparse spike 1) should be written in small letters. Please check the whole manuscript carefully.
“SS1” has been replaced with “ss1” throughout the paper.
- Authors have used different chemicals/reagents for a different purposes, please provide their catalog numbers where applicable.
Catalog numbers for chemicals/reagents has been added as suggested.
- Authors used a baking test with slight modifications but they did not mention these modifications.
The slight modifications can be found in “line 332: Specifically, a standard rapid mix test involving 50 g flour (14% moisture content) was conducted. There were two loaves of bread per flour sample”.
- Please add a conclusion section with strong future recommendations.
Line 246: In this study, we characterized two new Q alleles (Qs1 and Qc1-N8), and demonstrated that the negative correlation between GY and GPC of wheat can be broken by Qc1-N8 via a non-transgenic approach. Qc1-N8 synchronously increases the wheat GY and GPC, and a specific molecular marker was developed to facilitate its use in breeding. It is possible to further increase both GY and GPC by progressively optimizing of the Q gene.